# SMRT and Illumina RNA Sequencing and Characterization of a Key *NAC* Gene *LoNAC29* during the Flower Senescence in *Lilium oriental* ‘Siberia’

**DOI:** 10.3390/genes12060869

**Published:** 2021-06-06

**Authors:** Jing Luo, Ruirui Li, Xintong Xu, Hairui Niu, Yujie Zhang, Caiyun Wang

**Affiliations:** 1Key Laboratory for Biology of Horticultural Plants, Ministry of Education, College of Horticulture and Forestry Sciences, Huazhong Agricultural University, Wuhan 430070, China; ljcau@mail.hzau.edu.cn (J.L.); lrr510819@163.com (R.L.); xintong@webmail.hzau.edu.cn (X.X.); niuhairui0115@gmail.com (H.N.); zyjj328@163.com (Y.Z.); 2Key Laboratory of Urban Agriculture in Central China, Ministry of Agriculture, Wuhan 430070, China

**Keywords:** *Lilium* spp., flower senescence, SMRT sequencing, senescence-associated gene, NAC

## Abstract

Lily (*Lilium* spp.) is an important cut flower around the world. Flower senescence in lilies is characterized by the wilting and abscission of tepals, which results in a decrease in flower quality and huge economic loss. However, the mechanism underlying flower senescence in lilies is largely unknown. In this study, single-molecule, real-time (SMRT) and Illumina sequencing were carried out in *L**. oriental* ‘Siberia’. Sequencing yielded 73,218 non-redundant transcripts, with an N50 of 3792 bp. These data were further integrated with three published transcriptomes through cogent analysis, which yielded 62,960 transcripts, with an increase in N50 of 3935 bp. Analysis of differentially expressed genes showed that 319 transcription factors were highly upregulated during flower senescence. The expression of twelve *NAC* genes and eleven senescence-associated genes (*SAGs*) showed that *LoNAC29* and *LoSAG39* were highly expressed in senescent flowers. Transient overexpression of *LoNAC29* and *LoSAG39* in tepals of lily notably accelerated flower senescence, and the promoter activity of *LoSAG39* was strongly induced by *LoNAC29*. This work supported new evidence for the molecular mechanism of flower senescence and provided better sequence data for further study in lilies.

## 1. Introduction

Senescence is a natural process in which organs lose the capacity for cell division and gradually die. During vegetative growth in plants, under nutrient deficiency or moderate stress, old leaves initiate the process of senescence, allowing plants to use limited nutrients to sustain young leaves or to resist environmental stress [1]. During reproductive growth, plants use flowers to attract insects or birds for pollination. Once pollination is complete, ethylene is synthesized in the ovary, causing the flowers to senesce rapidly and deliver nutrients to seeds or other organs, as well as to maintain normal plant growth [2].

Plants can be divided into ethylene-sensitive and -insensitive types [3]. For ethylene-sensitive flowers, ethylene plays an important role in flower senescence. In tulips (*Tulipa gesneriana*), flower senescence is accompanied by large amounts of ethylene synthesis. *ACS* is a key gene in ethylene synthesis, which encodes 1-aminocyclopropane-1-carboxylate (ACC) synthase. Silencing of *ACS* significantly delays tepal senescence in tulips [4]. Most of the lily cultivars, however, are less sensitive or insensitive to ethylene, so ethylene inhibitor silver thiosulfate (STS) or 1-methylcyclopropene (1-MCP) has little effect on prolonging the flower life of cut lily *L. oriental* ‘Stargazer’ [5].

In addition to ethylene, hormones such as abscisic acid (ABA) also play a role in promoting flower senescence. ABA promotes flower senescence in both ethylene-sensitive and -insensitive plants. The increase in ABA content in lily flowers is closely related to the senescence of tepals [6]. This phenomenon is also found in other ethylene-insensitive flowers such as gladiolus (*Gladiolus palustris*) [7].

Downstream of the hormone signal, many transcription factors, and functional genes are involved in the regulation of flower senescence. NAC (for no apical meristem (NAM), Arabidopsis transcription activation factor (ATAF), and cup-shaped cotyledon (CUC)) is a large transcription factor family in plants. It was reported that many *NAC* genes were involved in leaf senescence; inhibition of a *NAC* gene *AtNAP* significantly delays leaf senescence [8], which is important for increasing crop yields and alleviating the food crisis. In morning glory (*Ipomoea nil*), elimination of one *NAC* gene, *EPHEMERAL1*, by CRISPR/Cas9 significantly delays flower senescence [9]. In addition, several other transcription factors from the MYB and homeodomain-leucine Zipper (HD-Zip) families have been reported to play important roles in flower senescence of rose (*Rosa hybrida*) [10,11,12]. Downstream of transcription factors, many senescence-associated genes are directly involved in the regulation of flower senescence, including degradation of proteins and lipids, defense, transport, and destruction of the cell wall [13]. In rose (*R**. hybrida*), RhERF4 regulates pectin metabolism via direct regulation of the pectin degradation-related gene *β-GALACTOSIDASE 1* (*RhBGLA1*), which leads to petal abscission [14]. However, the regulatory mechanism underlying flower senescence is still largely unknown.

In order to study the mechanism of transcriptional regulation, transcriptome sequencing is widely used to screen key genes. Through Illumina sequencing, several transcription factors, such as Ethylene Responsive Factor (ERF), Auxin Responsive Factor (ARF), basic Helix-Loop-Helix (bHLH), HD-Zip, and MADS-box, were found to be related to flower senescence in petunia (*Petunia hybrida*), and their function was verified by virus-induced gene silencing (VIGS) [15]. In lilies, Illumina sequencing is also widely used to screen key genes involved in flower aroma synthesis [16,17], vernalization of bulbs [18], and flower color formation [19]. Illumina sequencing is efficient for obtaining gene expression patterns and for screening for important differentially expressed genes. However, for some species without a reference genome, Illumina sequencing makes it difficult to obtain the full length of the genes, which is a big obstacle for gene cloning and further study. With the development of sequencing technology, the third-generation sequencing technology, represented by Pacbio SMRT, has developed rapidly, which is of great value in obtaining higher quality gene sequences and is used in studying alternative splicing of mRNA [20].

Lily (*L*. spp.) is a perennial bulb flower of the *Lilium* genus, and one of the top four cut flowers in the world [17]. The vase life of lily cut flowers is relatively short, because senescence seriously affects its ornamental quality, leading to great economic loss. In order to study the regulatory mechanism underlying flower senescence in *L. oriental* ‘Siberia’, in this study, SMRT and Illumina sequencing were carried out and, through Cogent analysis, the sequencing data were integrated with three published sets of transcriptome data in lily. Hence, a better version of sequence data was generated. Additionally, many differentially expressed genes related to flower senescence, color, and scent were obtained. LoNAC29 was found to be a key transcription factor involved in flower senescence, partly by regulating the expression of a senescence-associated gene *LoSAG39*. This work provided a better understanding of flower senescence and established a good basis for further study in lilies (*L.* spp.).

## 2. Materials and Methods

### 2.1. Plant Materials

Lily (*L. oriental* ‘Siberia’) was grown in a greenhouse located in the campus of Huazhong Agricultural University in the spring. The room temperature ranged from 15 to 25 °C. The air humidity was kept at 60–70%. The development stages of the lily flower were defined according to Shi et al. [17], with minor modification: stage 1, flower bud grows to the largest size, color of tepals does not turn white; stage 2, early opening, the upper one third of the tepals were opened with an angle of 30 degrees; stage 3, full opening, the tepals extend to the maximum extent; stage 4, flowers are senescent, the tepals are wilting (Figure 1). Flowers of different stages were sampled at the same time, with three biological replicates for each stage. The samples were quickly wrapped in foil, frozen in liquid nitrogen, transported to the lab, and stored in a −80 °C freezer. These samples were prepared for SMRT and Illumina sequencing.

As for transient overexpression in lily tepals, cut flowers of lily (*L. oriental* ‘Siberia’) were bought from Nanhu flower market and transported to the laboratory within one hour. The flower stems were cut to a length of 30 cm and the leaves at the stem base were removed; the flowers were then placed in vases with distilled water.

Potted flower of lily (*L. asiatic* ‘Orange Matrix’) was bought from Nanhu flower market before the flowers were opened and further cultivated in the greenhouse. Growth conditions were the same as with lily (*L. oriental* ‘Siberia’). The flower opening process was also divided into four stages as above (Appendix A). Flowers of different stages were sampled at the same time, with three biological replicates for each stage. The samples were quickly wrapped in foil, frozen in liquid nitrogen, transported to the lab, and stored in a −80 °C freezer. These samples were prepared for gene expression analysis.

### 2.2. RNA Extraction, Illumina Sequencing, and SMRT Sequencing

RNA was extracted from lily samples with the hot borate method [21]. The concentration, purity, and integrity of RNA was checked with Agilent 2100 (Agilent, Palo Alto, CA, USA), Nanodrop 2000 (Thermo Fisher Scientific, Waltham, MA, USA), and agarose gel electrophoresis. After that, Illumina sequencing was carried out on Illumina HiSeq 2500 platform. Based on the quality of RNA, among the three biological repeats, the best one of each stage was chosen, and mixed to form one sample for SMRT sequencing with the PacBio RSII platform. Sequencing was finished by Novogene Bio Technology Co., Ltd. in Beijing, China. For raw data of SMRT sequencing, SMRTlink V6.0 (parameters: —minLength 50; —maxLength 15,000; —minPasses 2; —minPredictedAccuracy 0.8; —minZScore −9999; —maxDropFraction 0.8; —min_seq_len 200; —minReadScore 0.65) was used to remove adapters and low-quality reads, thus obtaining the subreads. After self-correction between the subreads, circular consensus sequences (CCSs) were obtained. The full-length non-chimeric (FLNC) and non-full-length (NFL) sequences were divided, according to whether the sequences contained 5′ and 3′ adapters and poly(A) tails. Cluster consensus sequences were obtained by clustering FLNC and sequences from the same transcript using the iterative clustering for error correction (ICE) algorithm [22]. Finally, the NFL sequences were used to polish the consistent sequence with the Arrow algorithm (https://downloads.pacbcloud.com/public/software/installers/smrtlink_9.0.0.92188.zip, accessed on 10 May 2019) [23], and the high-quality polished consensus sequence was obtained for further analysis. Illumina sequencing data were used for further proofing the polished consensus sequence with LoRDEC software (parameters: -k 23, -s 3) [24], followed by redundancy removal with CD-HIT software with the following parameters: -c 0.99, –G 0, –aL 0.00, –aS 0.99, –AS 30, -M 0, –d 0, –p 1 [25]. Next, the non-redundant transcripts were generated and further annotated with seven public databases: NCBI non-redundant protein sequences (NR), NCBI non-redundant nucleotide sequences (NT), Protein family (Pfam), Clusters of Orthologous Groups of proteins (KOG/COG), a manually annotated and reviewed protein sequence database (Swiss-Prot), KEGG Ortholog database (KO), and Gene Ontology (GO).

### 2.3. Identification of Differentially Expressed Genes (DEGs)

The clean data in each sample were mapped to the reference sequence (non-redundant transcripts) using the Bowtie2 (parameters: -q, —phred33, —sensitive, —dpad 0, —gbar 99,999,999, —mp 1,1, —np 1, —score-min L,0,−0.1, —-I 1, —-X 1000, —no-mixed, —no-discordant, —-p 8, —-k 30) and RNA-Seq tools of Expectation-Maximization (RSEM) software (parameters: —phred33, —-quals, —forward-prob 0.5, —time), and the read counts were generated. Based on the read counts, the DEGs were analyzed using DESeq [26] with the fold change >2.0, and padj < 0.05. Kyoto Encyclopedia of Genes and Genomes (KEGG) analysis of DEGs were carried out with KOBAS (v2.0.12) software, padj < 0.05.

### 2.4. Coding GENome Reconstruction Tool (Cogent) Analysis

Cogent analysis was carried out with default parameters, according to the method described in [27]. The k-mer profiles of non-redundant transcripts from our SMRT sequencing (group B) and three published transcriptomes based on Illumina sequencing in lilies [16,17,18] (group A) were created, pairwise distances of transcripts were calculated, and similar transcripts were clustered into families based on their k-mer values. For each family, similar transcripts were reconstructed into one new transcript by using a De Bruijn graph method. In the De Bruijn graph, Cogent analysis was carried out based on the coverage and identity of the overlap. For example, An from group A and Bm from group B were clustered into the same family. If the identity <95%, return Bm (category one) and, if the coverage >95%, the identity >95%, and the length of An < Bm, return Bm (category two). Category one and category two were merged and named Pacbio.un.blast. If the coverage >95%, the identity >95%, and the length of An > Bm, the sequence of An was assigned to Bm (category three, named Final); if the coverage <95% and the identity >95%, the sequence of An and Bm were reconstructed into one new transcript (category four, named Cogent) (Figure 2).

### 2.5. TransDecoder Analysis

TransDecoder analysis (default parameters) was carried out according to the method described in [28]. Firstly, a minimum length of opening reading frame (ORF) (default 100 aa) was identified in the transcripts, to further maximize the sensitivity of the functional ORF. The ORF was compared with the UniProt and PFAM databases to identify the common protein domain. Finally, the final prediction was scored based on the alignment in the two databases. Based on the TransDecoder analysis, the transcripts were classified into four categories: complete, 5prime_partial, 3prime_partial, and internal. Complete means the sequence contains the complete opening reading frame (ORF), 5prime_partial means the 5′ end of the sequence was missing, 3prime_partial means the 3′ end was missing, and internal means both 5′ and 3′ parts were missing. For genes which were predicted to have multiple ORFs, the one with the highest score was reserved. For genes which were predicted as complete, 5prime_partial, or 3prime_partial, if the score was less than −20, they were merged with the internal category and renamed ‘internal and others’.

### 2.6. Quantitative Real-Time (qRT)-PCR Analysis

For qRT-PCR, the cDNA template was synthesized with 1 μg RNA by using cDNA Synthesis SuperMix (AE311-03, TransGen Biotech, Beijing, China) according to the manufacturer’s instructions. The primers used for qRT-PCR were designed based on the SMRT sequencing data in lily (*L. oriental* ‘Siberia’), as listed in Appendix A. *ACTIN* in *L. oriental* ‘Siberia’ (*LoA**CTIN*) was used as the internal control gene, according to [29]. All reactions were performed with three biological replicates on LightCycler 96 (Roche, Basel, Switzerland). The data were calculated with 2^−ΔΔCT^ method [30]. Two-sided Student’s *t*-tests (* *p* < 0.05; ** *p* < 0.01) were used for statistical analysis.

### 2.7. Gene Cloning and Vector Construction

The sequences of *LoNAC29* and *LoSAG39* were extracted from the SMRT sequencing data, and the promoter of *LoSAG39* was obtained by using fusion primer and nested integrated (FPNI)-PCR [31]. The sequences were corrected by using high-fidelity Phusion DNA Polymerase (Thermo Fisher Scientific, Waltham, MA, USA).

For overexpression (OE) of *LoNAC29* and *LoSAG39*, the ORF carrying the proper restriction site was amplified by PCR, digested, and inserted into the pSuper1300 vector, which was derived from pCAMBIA1300 (laboratory of Dr. Zhizhong Gong, China Agricultural University). For promoter activity analysis, the promoter of *LoSAG39* harboring the appropriate restriction site was digested and inserted into the PBI121 vector.

### 2.8. Transient Overexpression in Lily Tepals

Overexpression of *LoNAC29* and *LoSAG39* in lily tepals was performed as previously described [12] with minor modification. The pSuper-*LoNAC29*, pSuper-*LoSAG39* or pSuper (empty vector) were transformed into *Agrobacterium tumefaciens* GV3101 separately. After culture overnight, the cells were harvested and resuspended in infiltration buffer (10 mM 2-morpholinoethanesulfonic acid [MES], 10 mM MgCl_2_, 20 μM acetosyringone [As], pH 5.6). OD_600_
_nm_ was adjusted to 0.8 and incubated in the dark without shaking for two hours. Lily tepal discs with a diameter of 12 mm were obtained using a hole punch and infiltrated with *A. tumefaciens* GV3101 cells under a vacuum (0.6 atm). The discs were then washed in deionized water and cultured on a Petri dish with wet filter paper in the dark at 8 °C for three days. After that, the Petri dish containing the discs were taken out, and the time point was defined as 0 day. Thereafter, the discs were cultured in Petri dishes at 23 °C and 60% relative humidity under 5000 lux light.

### 2.9. Transient Transactivation in Tobacco (Nicotiana benthamiana)

The effector pSuper1300 empty vector, or *pSuper::LoNAC29*, and reporter PBI121 empty vector, or PBI121 harboring the promoter of *LoSAG39*, were transferred into *A. tumefaciens* GV3101. After overnight culture, cells were collected and resuspended in the infiltration solution as above, and the density was adjusted to OD_600 nm_ = 1.6. Agrobacterium carrying the effector or reporter were mixed with the ratio 1:1 (*v*/*v*) and incubated in dark at 23 °C without shaking for two hours. Agrobacterium were then injected into the back of *N. benthamiana* leaves using a 1 mL syringe (needle was removed before injection). After cultivation in the dark at 23 °C and a relative humidity of 40–60% for three days, the leaves of treated *N. benthamiana* were collected, immediately frozen in liquid nitrogen, and stored at −80 °C.

### 2.10. Determination of β-Glucuronidase (GUS) Activity

Determination of GUS activity was performed according to [32]. Briefly, treated leaves of *N. benthamiana* were ground in liquid nitrogen, and the soluble protein was extracted with extraction buffer. To measure the GUS activity, 4-Methylumbelliferyl β-d-glucuronide hydrate (4-MUG) was used as a substrate. The fluorescence of the 4-MU product was measured with a fluorescence spectrophotometer (F-4500; Hitachi, Tokyo, Japan) at 365 nm excitation and 455 nm emission. The GUS activity was defined as the amount of 4-MU produced per minute (unit: μM 4-MU mg^−1^ (protein) min^−1^).

## 3. Results

### 3.1. SMRT Sequencing of Lily during Flower Opening

Through PacBio Sequel platform, 454,080 polymerase reads were obtained. The total data were 23.26 G nucleotides, and the average length was 51,217 bp. After removing the adapter and the original reads whose length is less than 50 bp, 13,782,679 subreads were obtained, with an average length and N50 of 1613 bp and 2281 bp, respectively. After checking the consistency of subreads going through the ZMW wells, 396,493 CCS (circular consensus sequence) sequences were obtained, with a mean length of 2397 bp and N50 of 3240 bp. Based on the 5′ primer, 3′ primer, and poly A, 310,912 FLNC sequences were found in CCS, with a proportion of 78.42%. After removing the redundancy, 73,218 non-redundant transcripts were obtained, with the total number of 203,392,716 nucleotides and N50 of 3792 bp (Table 1 and Appendix A). The result showed that 60,676 unigenes have at least one annotation, and 16,774 unigenes have annotations in all seven databases (Appendix A and Appendix A). Among the 56,644 unigenes annotated in NR database, 12,347 unigenes with top hits to *Elaeis guineensis* homolog genes, followed by *Phoenix dactylifera* (10,145 unigenes), *Ananas comosus* (3415 unigenes), *Musa acuminata* (3277 unigenes), *Asparagus officinalis* (2417 unigenes), and *Anthurium amnicola* (2350 unigenes) (Appendix A).

### 3.2. Cogent and TransDecoder Analysis of SMRT Sequencing Data

Several Illumina sequencing data in lily were published recently [16,17,18], which supplied very important information for the study of lily, since its genome sequencing has not been reported yet. In order to make good use of the published transcriptome data to optimize the SMRT sequencing data, Cogent analysis was conducted. After Cogent analysis, 36,471 Pacbio.un.blast sequences, 16,167 Final sequences and 10,322 Cogent sequences were obtained. The total was 62,960, composed of 186,674,771 nucleotides, and the N50 was 3935 bp (Table 1 and Appendix A). The quality of data was much higher than the original SMRT sequencing data and published second-generation transcriptome data.

The integrated sequencing data were further analyzed with TransDecoder. The result showed that in the original SMRT sequencing data, 43,998 sequences (60%) had the full length, 12,938 sequences (18%) were 5prime_partial, 699 sequences (1%) were 3prime_partial, and 15,583 sequences (21%) were internal and others which were missing both the 5′ and 3′ ends or the score of alignment in UniProt and PFAM databases were lower than −20. After Cogent analysis, 42,027 sequences (67%) were full length, 7774 (12%) and 548 (1%) were missing their 5′ and 3′ end, respectively, and 12,611 (20%) were internal and others (Table 2).

### 3.3. Analysis of DEG and Transcription Factor (TF) during Flower Opening

By using DESeq software, DEGs (fold change > 2, padj < 0.05) were determined between different opening stages in lily. Stage 1 versus stage 4 pairwise had the most DEG genes, 5071 genes were upregulated and 10,172 genes were downregulated. In contrast, stage 2 versus stage 3 had the least DEG genes (Appendix A). The DEGs between different stages were gathered, and a total of 20,650 DEGs were obtained (Appendix A). Clustering of 20,650 DEGs revealed that 665 genes were DEG in all four pairwises, including stage 1 versus stage 3, stage 1 versus stage 4, stage 2 versus stage 4, and stage 3 versus stage 4, and, compared with other stages, 5135 genes had a high expression level in stage 4, while 3554 genes had a low expression level in stage 4 (Figure 3 and Appendix A). Further, 20,650 DEGs were clustered into 12 subclusters. In subcluster 5, subcluster 7, and subcluster 8, the expression of DEGs was much higher in stage 4 than other stages. Meanwhile, in subcluster 1, subcluster 2, and subcluster 11, the DEGs had a much lower expression in stage 4 than in other stages (Appendix A).

Through SMRT and Illumina sequencing, 2081 TF and transcription regulators were obtained. Of those, 319 genes were highly expressed at stage 4 (senescence), while 511 genes were notably downregulated with senescence of the flowers (Appendix A). Most of the differently expressed TF came from the NAC, WRKY, bHLH, C2H2 Zinc finger, MADS, and MYB families. For example, in the NAC family, 28 of 75 genes were notably upregulated and 8 were remarkably downregulated at stage 4 compared with other stages (Figure 3 and Appendix A). Based on sequence alignment, we found that 13 *NAC* unigenes, as well as 1 *HB-HD-ZIP* unigene and 1 *ERF* unigene, were homologous to several published senescence-associated genes, including *ANAC029*(*AtNAP*) [8], *ANAC046* [33], *ANAC032* [34], *ANAC087* [35], *ANAC022*(*NAC1*) [36], *ANAC002*(*ATAF1*) [37], *Ipomoea nil EPHEMERAL 1* (*NAC092*) [9], *RhHB1* [10], and *RhERF113* [11] (Appendix A), and their expression was significantly upregulated during flower senescence, suggesting that they might also be involved in flower senescence of lilies.

### 3.4. Analysis of Color- and Scent-Related Genes during Flower Opening

During stage 1 of the flower opening in *L. oriental* ‘Siberia’, the tepals were green, and with the opening of flowers, they gradually turned white (Figure 1). The chlorophyll content was more than two times higher in stage 1 than in other stages, and there was no notable difference between stage 2, stage 3, and stage 4 (Appendix A). Chlorophyll metabolism was tightly related to photosynthesis. The KEGG pathway analysis showed photosynthesis and the chlorophyll metabolism pathway were enriched in stage 1 versus stage 2, stage 1 versus stage 3, and stage 1 versus stage 4. Next, 29 genes related to chlorophyll metabolism and 51 genes related to photosynthesis were downregulated in stage 2, stage 3, and stage 4 compared with stage 1 (Appendix A). Most of the genes related to chlorophyll metabolism were classified into subcluster 1, subcluster 2, and subcluster 11. Additionally, their expression was highly correlated with 114 transcription factors (Appendix A and Appendix A), whether these transcription factors were involved in regulation of chlorophyll metabolism will be checked in the further studies.

Flower opening is often accompanied with a release of flower scent. Terpenoids are major components in the flower scent of *L. oriental* ‘Siberia’, and the release of flower scent in lily mainly occurs in the full-flowering period (stage 3) [17]. KEGG pathway analysis indicated that the terpenoid backbone biosynthesis pathway was enriched in stage 1 versus stage 2 and stage 1 versus stage 3, and the expression of 54 genes related to terpenoid synthesis increased with flower opening, peaked at stage 3, and dropped at stage 4 (Appendix A). There were 30 genes related to terpenoid biosynthesis that were enriched in subcluster 9 and subcluster 10. Their expression pattern was in accordance with 28 transcription factors (Appendix A and Appendix A), and some of them were being verified whether they participated in terpenoid synthesis.

### 3.5. Analysis of Senescence-Associated Genes during Flower Opening

Visible flower senescence occurred in stage 4, and DEG analysis indicated that, compared with the other three stages, 5135 genes were strongly induced in stage 4 (Appendix A). Among them, eleven genes, annotated as senescence-specific cysteine protease, KDEL cysteine protease, or senescence-associated protein, were selected (Appendix A). qRT-PCR analysis showed that the expression of the 11 *SAG* genes sharply increased in stage 4 (Figure 4). Among these 11 *SAG* genes, the expression of *SAG1* is more obvious.

The expression of the 11 *SAG* genes was also checked in different organs, including root, stem, mature leaf, flower, and senescent leaf. Relative to the expression in the root, the expression of *SAG1*, *SAG5,* and *SAG8* was extremely high in the flower (Appendix A). When compared with the mature leaf, the expression of *SAG5*, *SAG6*, and *SAG8* was more than 30,000 times that of the senescent leaf (Appendix A), which suggests that these genes might be also involved in leaf senescence. Since the expression of *SAG1* was more obvious in flowers undergoing senescence, the full length of *SAG1* was cloned, and phylogenetic analysis showed that the protein sequence of SAG1 had high homology with the SAG39 proteins in multiple species, so the gene was named *LoSAG39* (Appendix A, Genbank accession no. MW520943).

Transient overexpression of *LoSAG39* was carried out in lily tepal discs. The results showed that, after four days, in the super1300 vector control samples, about half of the tepal discs were senescent, while in *LoSAG39* OE samples, nearly 100% of the tepal discs showed the senescent phenotype. qRT-PCR analysis indicated that the expression of *LoSAG39* remarkably increased, compared with control samples. The ion leaking rate also showed that the damage of the cell membrane system is more serious in *LoSAG39* OE samples (Figure 5).

### 3.6. LoNAC29 Accelerated Flower Senescence Partly by Activating Several SAGs

Analysis of TF showed that 37.33% (28/75) *NAC* genes were upregulated during stage 4, when flowers were going through aging. We selected 12 *NAC* genes for further analysis (Figure 3 and Appendix A). The expression of the 12 genes was checked in stage 1 to stage 4 of *L. oriental* ‘Siberia’ flowers, the results indicated that most of them, including *NAC2*, *NAC3*, *NAC4*, *NAC5*, and *NAC10*, were sharply induced in stage 4 (Figure 6). To confirm whether the high expression of *NAC* genes in stage 4 was related to flower senescence, the expression of *NAC2* and *NAC4* were also checked in different opening stages of lily (*L. Asiatic* ‘Orange Matrix’) flowers. The result showed that the expression of *NAC2* rapidly increased more than 1500 times in stage 3 and 2000 times in stage 4, compared with stage 1. However, the change in *NAC4* was not as strong as *NAC2*. Additionally, both *NAC2* and *NAC4* were induced by ABA treatment (Appendix A).

The full length of *NAC2* was cloned, and phylogenetic analysis showed that it had high homology with AtNAC29 in *Arabidopsis thaliana*, so the gene was named *LoNAC29* (Appendix A, Genbank accession no. MW548579). Overexpression of *LoNAC29* significantly accelerated the senescence of lily flowers. In *LoNAC29* OE samples, the expression of *LoNAC29* was significantly upregulated compared with pSuper vector control, and the ion leakage rate was also much higher in *LoNAC29* OE samples, which suggested that overexpression of *LoNAC29* resulted in more serious flower senescence (Figure 7).

In *LoNAC29* OE samples, the expression of *LoSAG39*, *SAG2,* and *SAG7* was notably upregulated (Figure 8A). The promoter of *LoSAG39* was cloned by using FPNI-PCR, and the promoter fragment with a length of 734 bp was obtained. Transient transactivation assay was carried out in the leaves of *N. benthamiana*, and the result showed that the promoter activity of *LoSAG39* was also significantly induced by LoNAC29 (Figure 8B).

## 4. Discussion

The flower is an important organ of angiosperm that evolved from leaves [38]. In cut flowers, the supply of water and nutrients are cut off after harvest, and how to maintain flower quality is a big problem. The cut lily flowers cannot tolerate a long period of storage and transportation, and the life of the flowers is relatively short. Delaying the senescence of the cut lily flowers is important for improving the ornamental value and prolonging the vase life of cut lily flowers.

### 4.1. Integration of SMRT and Illumina Sequencing Is an Efficient Approach for Study in Plants without a Reference Genome

Ploidy of lilies (*L.* spp.) is very complex, as most of the cultivars are diploid or tetraploid, and some are triploid. *L. oriental* ‘Siberia’ is diploid (2*n* = 2x = 24) [39]. The genome of lilies has not been reported yet, but it is estimated to be 36 Gb in size, since it contains a large number of repeats [40]. Previous studies on lilies have focused on the formation of floral color and fragrance, heat tolerance, and bulb dormancy. Due to the lack of a reference genome, many studies used Illumina sequencing to obtain gene sequences and study the function of candidate genes [18,19]. However, in Illumina sequencing, transcripts are typically cut into multiple pieces. Due to the short length of reads, the assembled transcripts were often not complete, thereby impeding the cloning of target genes and further study of gene function.

The third-generation sequencing technology, represented by Pacbio SMRT, developed rapidly and played an important role in biological research [20]. In this study, we used SMRT and Illumina sequencing to study the molecular mechanism of flower senescence in lily. A total of 73,218 transcripts were obtained, and the N50 was 3792 bp. In contrast, in the published transcriptome of lily based on Illumina sequencing, the N50 was 981 bp [16], 1038 bp [17], and 1443 bp [18] (Appendix A). Further, we combined our SMRT sequencing data with the published Illumina sequencing data for Cogent analysis. We obtained 62,960 transcripts, and the N50 was further extended to 3935 bp, which was much better than the published transcriptome data of lily (*L.* spp.). Among the 62,960 transcripts, 42,027 (67% of the sequences) were predicted to have the full length by TransDecoder analysis, which was also better than our original SMRT sequencing data (Table 2).

### 4.2. Senescence-Associated Genes and Key Regulator during Flower Senescence in L. Oriental ‘Siberia’

In lilies (*L.* spp.), the senescence of flowers is presented as the wilting and abscission of tepals, accompanied by the yellowing and falling off of leaves. With regard to plant senescence, previous studies focused on leaf senescence—in particular, the synthesis and regulation of chlorophyll in leaves [41]. The study of leaf senescence is of great significance for increasing crop yield. Flowers and leaves have different functions; however, the mechanism underlying senescence of leaves and flowers is similar. The regulation of hormone signals such as ethylene and ABA, the accumulation of reactive oxygen species (ROS), and the degradation of biomacromolecules also occur in flower and leaf senescence [3,42]. In this study, the expression analysis of 11 senescence-related genes in lily showed that *SAG1* (*LoSAG39*), *SAG5*, *SAG6*, *SAG7*, and *SAG8* had a very high expression level in senescent flowers and leaves, suggesting that they may be involved in both leaf and flower senescence (Figure 4 and Appendix A). Based on sequence alignment on TAIR (https://www.arabidopsis.org/, accessed on 25 May 2021), *SAG1* (*LoSAG39*), *SAG2,* and *SAG7* were homologous to the important senescence marker gene *SAG12* in Arabidopsis, which encodes a cysteine protease [8]. SAG3, SAG10, and SAG11 were homologous to AT1G11190, a bifunctional nuclease that acts on nucleic acid degradation, and the gene expression was extremely high in petal and stamen during flower senescence of Arabidopsis. Additionally, the sequence of KDEL cysteine protease unigenes *SAG4*, *SAG5*, *SAG6*, *SAG8*, and *SAG9* were similar to AT5G50260 (*AtCEP1*), which was involved in pathogen defense, and *atcep1* knockout mutants showed enhanced susceptibility to powdery mildew caused by the biotrophic ascomycete *Erysiphe cruciferarum* [43]. This work suggested that the 11 selected *SAG* unigenes might be involved in protein/nucleic acid degradation and pathogen defense during flower senescence in lilies. In 319 differentially expressed genes which were highly expressed at stage 4, except for the 11 selected *SAG* unigenes, some unigenes were annotated as gibberellin 2-β-dioxygenase (GA2ox, i1_LQ_lily14_c66443/f1p8/1251, i1_LQ_lily14 _c95653/f1p4/1051 and i1_LQ_lily14_c88156/f1p3/1464), cytokinin dehydrogenase (CKX, i1_LQ_lily14_c40871/f1p1/1691, i0_HQ_lily14_c20393/f3p0/974, i1_LQ_lily14_c13708 /f1p1/1454), cis-zeatin O-glucosyltransferase (i1_HQ_lily14_c101975/f4p6/1575, and i1_HQ_lily14_c97578/f5p6/1552), and 9-cis-epoxycarotenoid dioxygenase (NCED, i1_LQ_lily14_c6047/f1p0/1704) (Appendix A), suggesting that hormone synthesis and signaling were widely involved in flower senescence of lilies. In addition, there were also several unigenes, including i0_LQ_lily14_c45469/f1p0/532, i1_LQ_lily14_c84559 /f1p5/1076, i2_LQ_lily14_c33901/f1p0/2035, i1_LQ_lily14_c68506/f1p1/1482, i2_LQ_lily14 _c42411/f1p0/2962, i0_LQ_lily14_c7791/f1p0/914, i1_LQ_lily14_c84085/f1p4/1556, and i3_LQ_lily14_c38942/f1p18/3354 were homologous to E3 ubiquitin-protein ligase and F-box (Appendix A), which were reported to play important roles in protein degradation [44]. Whether these genes mediate flower senescence in lilies needs further investigation.

In terms of the regulatory mechanism underlying flower senescence, it was reported that an ethylene-induced HD ZIP transcription factor RhHB1 inhibits *GA20ox1*, a key gene for gibberellin synthesis, thus accelerating flower senescence [10]. On the other hand, RhERF113 promotes the synthesis of cytokinin and thus delays the senescence of rose flowers [11]. In this study, 2081 transcription factor genes were screened by SMRT and Illumina sequencing, and 319 of them were significantly upregulated at stage 4, compared with the other three stages. Among these upregulated transcription factors, many genes come from the NAC, WRKY, bHLH, and C2H2 Zinc finger families. Their expression displayed more significant changes (Figure 3 and Appendix A).

Since the discovery of a *NAC* gene *AtNAP* (*ANAC029*) that plays a key role in Arabidopsis leaf senescence [8], many studies have focused on the identification of key *NAC* genes in other species [45]. It was found that *NAC* genes promoted leaf senescence partly through inducing chlorophyll degradation [33], ABA synthesis [46], or ROS accumulation [47].

Some studies have also shown that the *NAC* gene regulates flower senescence in Japanese morning glory (*Ipomoea nil* cv. Violet). Knock-out of *NAC* gene *EPHEMERAL 1* significantly delays senescence in Japanese morning glory (*Ipomoea nil*) [9]. However, the detailed regulatory mechanism of the *NAC* gene in flower senescence is not clear.

We searched the homologous genes in our SMRT sequencing data with published transcription factors which were reported to be involved in leaf or flower senescence and found 13 *NAC* unigenes, as well as 1 *HB-HD-ZIP* unigene and 1 *ERF* unigene were homologous to *ANAC029* (*AtNAP*) [8], *ANAC046* [33], *ANAC032* [34], *ANAC087* [35], *ANAC022*(*NAC1*) [36], *ANAC002* (ATAF1) [37], *Ipomoea nil EPHEMERAL 1* (*NAC092*) [9], *RhHB1* [10], and *RhERF113* [11] (Appendix A), and their expression was significantly upregulated at stage 4, suggesting that they might be important candidate transcription factors that regulate flower senescence in lilies. In addition, we confirmed that *LoNAC29* is a key regulator in the process of lily flower senescence, and OE of *LoNAC29* significantly promotes the senescence of lily tepals. In the *LoNAC29* OE samples, the expression of *LoSAG39*, *SAG2*, and *SAG7* was significantly increased, and LoNAC29 strongly induced the promoter activity of *LoSAG39* (Figure 8). *LoSAG39*, *SAG2*, and *SAG7* are cysteine proteases, which may play a role in protein degradation. Overexpression of *LoSAG39* also accelerated the senescence of lily tepals (Figure 5).

### 4.3. Changes of Flower Color and Scent Are Accompanied by Flower Senescence in L. oriental ‘Siberia’

The process of flower senescence in lily involves the degradation of proteins as well as chlorophyll degradation. Here, in *L. oriental* ‘Siberia’, the flower color is green at stage 1 and, with the opening of flowers, the tepals gradually turn white. White flowers are formed partly as a result of chlorophyll degradation [48] and partly as a result of genetic mutations or reduced expression levels of anthocyanin and carotenoid synthesis-related genes [49,50]. Similarly, in this study, we also found that chlorophyll content decreased with flower opening in *L. oriental* ‘Siberia’ and, accordingly, the expression level of chlorophyll synthesis-related genes and photosynthesis-related genes also decreased gradually (Appendix A).

Some studies have also reported the mechanism of chlorophyll metabolism and regulation in green flowers. For example, in chrysanthemum (*Chrysanthemum morifolium* Ramat.), a NAC transcription factor was found to mediate chlorophyll synthesis partly through direct activation of *HEMA1* and *CRD1* genes [51]. In *L.*
*oriental* ‘Siberia’, most of the genes involved in chlorophyll synthesis and photosynthesis were clustered into subcluster 1, subcluster 2, and subcluster 11. These clusters contained 114 transcription factors (Appendix A, Appendix A). In the process of flower opening, the decline of chlorophyll synthesis accompanies flower senescence, and whether these transcription factors were involved in chlorophyll synthesis or flower senescence needs further investigation. Metabolism of chlorophyll was tightly correlated with the accumulation of photosynthetic products, and carbohydrates, as energy substances, play an important role in delaying flower senescence [52]. It was speculated that preventing chlorophyll degradation, or promoting its synthesis, may be a new strategy for delaying flower senescence.

The flower opening of lilies (*L.*
*oriental* ‘Siberia’) is also accompanied by the release of flower scent. The main components of floral fragrance are terpenoids, benzenoids/phenylpropanoid, and derivatives of fatty acids. Terpenes are synthesized by the MEP and MVA pathway. The main aroma components in *L.*
*oriental* ‘Siberia’, cis-ocimene, linalool, and other terpenes, are released in large amounts at full bloom stage [18]. In this study, we also found that 68 MEP and MVA genes were expressed at full bloom stage (stage 3). At present, a few scant reports indicate that synthesis of terpene is regulated by MYC2 and MYB21 in freesia (*Freesia hybrida*) flowers [53] and ERF71 in the peel of sweet orange (*Citrus sinensis* Osbeck) [54]. Here, in cluster analysis, 28 transcription factors were found to be associated with the expression patterns of genes involved in terpene biosynthesis (Appendix A). Among them, there were five MYB unigenes, including i0_HQ_lily14_c35718/f4p7/940, i1_HQ_lily14_c50407/f3p4/1037, i1_LQ_lily14_c39458/f1p4/1091, i0_LQ_lily14_c41028/f1p0/945, and i1_HQ_lily14_c99358 /f4p4/1023; the function of related genes needs to be further studied. It was reported that the flower longevity is different between two groups of fragrant and non-fragrant roses, in fragrant varieties, such as *Rosa bourboniana*, the flower life is much shorter, compared with most *R. hybrida* varieties, which are non-fragrant and widely used as cut flowers [55]. Synthesis of aroma components consumes a lot of energy and, in many species of the palm family, much energy is allocated to scent production in order to attract insects pollinators [56]. In future studies, controlling the synthesis of aroma components might be a new way to reduce the energy consumption and prolong the flower longevity.

## 5. Conclusions

Overall, through this study, we obtained a high-quality version of transcripts and made new insights into flower senescence in lily (*L. oriental* ‘Siberia’), which provided a good foundation for the further study of gene cloning and related molecular mechanisms in lily (*L.* spp.). Additionally, we found that the key regulator LoNAC29 promotes lily flower senescence by inducing the expression of *LoSAG39*.

## Figures and Tables

**Figure 1 genes-12-00869-f001:**
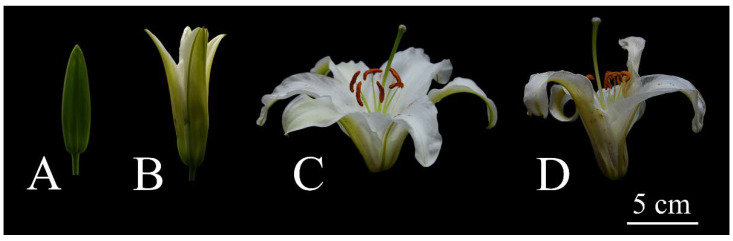
Flower opening stages in *Lilium oriental* ‘Siberia’. (**A**) Stage 1: flower bud grows to the largest size, color of tepals does not turn white; (**B**) stage 2: early opening, the upper one third of the tepals were opened with an angle of 30 degrees; (**C**) stage 3: full opening, the tepals extend to the maximum extent; (**D**) stage 4: flowers are senescent, the tepals are wilting.

**Figure 2 genes-12-00869-f002:**
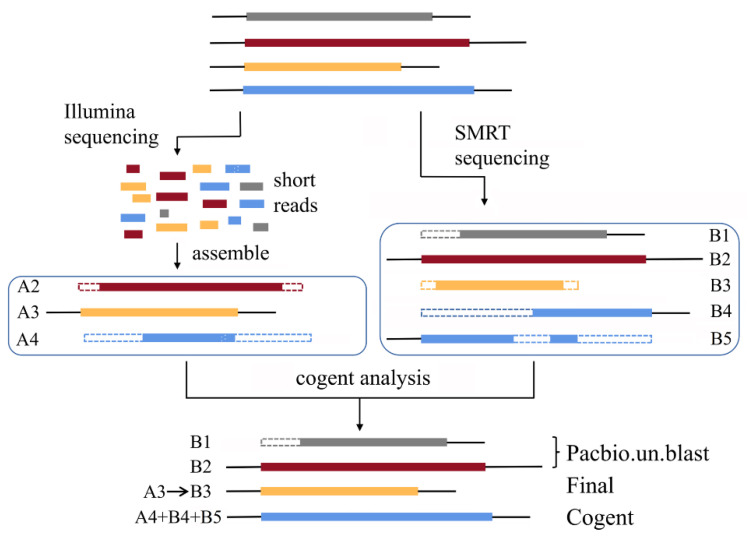
A scheme for cogent analysis of SMRT and Illumina sequencing data. Assembled Illumina sequencing data (group **A**) were compared with non-redundant transcripts coming from SMRT sequencing data (group **B**), and pairwise distances of transcripts were calculated. Based on the k-mer values, similar transcripts were clustered into the same family, and similar transcripts were reconstructed into one new transcript by using a De Bruijn graph method. In the De Bruijn graph, Cogent analysis was carried out based on the coverage and identity of the overlap. For example, An from group A and Bm from group B were clustered into the same family. If the identity <95%, return Bm (category one) and, if the coverage >95%, the identity >95%, and the length of An < Bm, return Bm (category two). Category one and category two were merged and named Pacbio.un.blast. If the coverage >95%, the identity >95%, and the length of An > Bm, the sequence of An was assigned to Bm (category three, named Final); if the coverage <95% and the identity >95%, the sequence of An and Bm were reconstructed into one new transcript (category four, named Cogent).

**Figure 3 genes-12-00869-f003:**
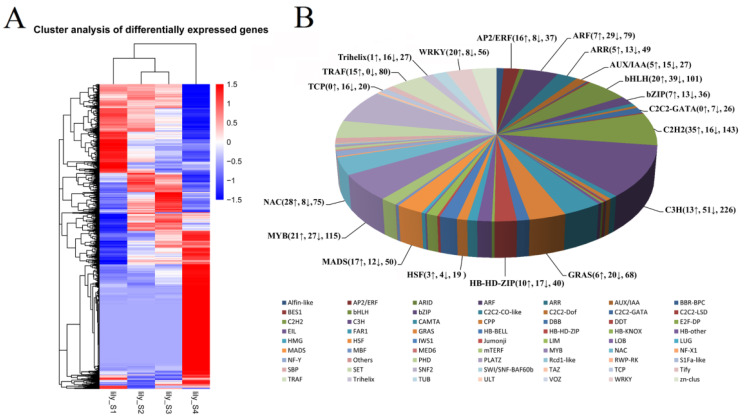
Cluster analysis of differentially expressed genes (DEGs) and distribution of transcription factors. (**A**) Heat map shows the expression of 20,650 DEGs in flowers of lily (*Lilium oriental* ‘Siberia’) at stage 1 to stage 4; (**B**) pie chart demonstrates the distribution of transcription factors. The numbers with up arrows and down arrows mean up- and downregulated genes at stage 4 compared with the other three stages, respectively, and the last number in brackets means the total number of genes obtained through SMRT sequencing.

**Figure 4 genes-12-00869-f004:**
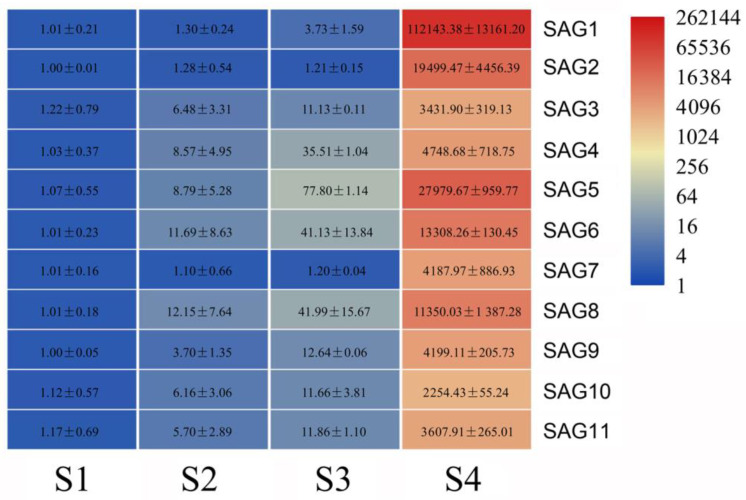
Expression of eleven senescence-associated genes during flower opening by qRT-PCR analysis. Different colors indicate the expression level of genes. The data represent the mean ± SE of three biological replicates. *A**CTIN* from lily (*Lilium oriental* ‘Siberia’) (*LoACTIN*) was used as the internal control.

**Figure 5 genes-12-00869-f005:**
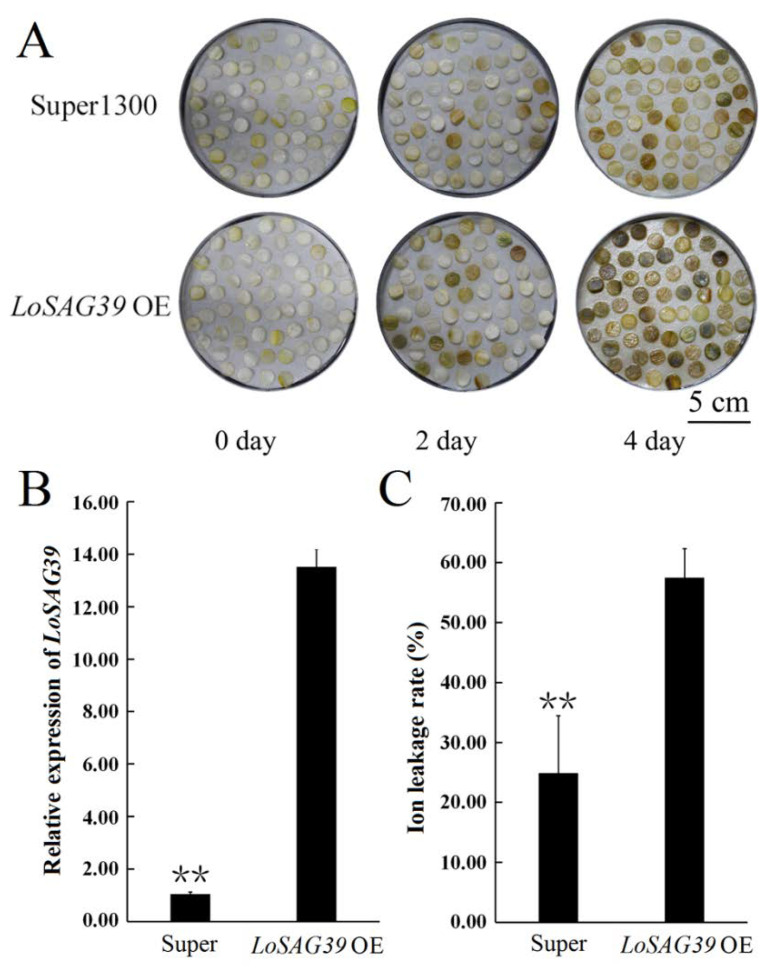
Overexpression of *Lilium oriental SAG39* (*LoSAG39*) in tepal discs of lily (*L. oriental* ‘Siberia’). (**A**)Phenotype of *LoSAG39* OE and super vector control samples. (**B**,**C**) Expression of *LoSAG39* and ion leakage rate in *LoSAG39* OE and super vector control samples. The data represent the mean ± SE of three biological replicates. ** *p* < 0.01 (Student’s *t*-test).

**Figure 6 genes-12-00869-f006:**
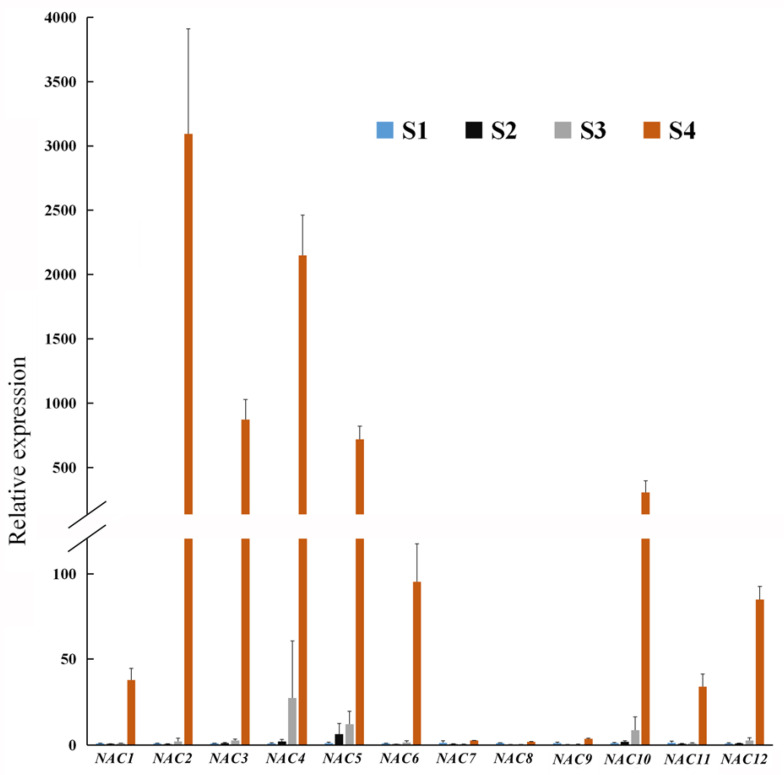
Expression of 12 *NAC* genes during flower opening by qRT-PCR analysis. The data represent the mean ± SE of three biological replicates. *LoACTIN* was used as the internal reference gene.

**Figure 7 genes-12-00869-f007:**
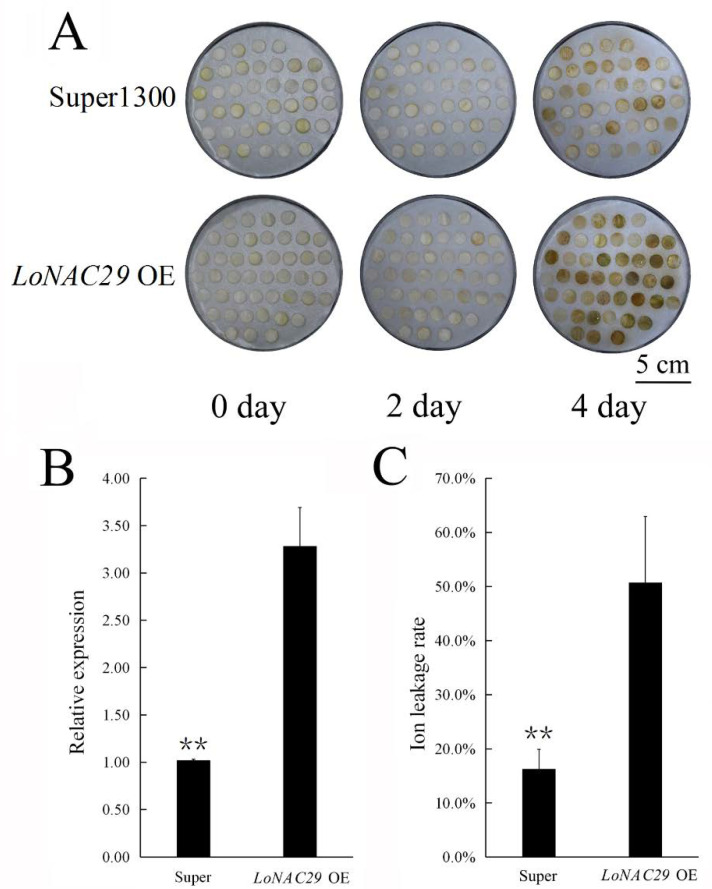
Overexpression of *Lilium oriental NAC29* (*LoNAC29*) in tepal discs of lily (*L**. oriental* ‘Siberia’). (**A**) Phenotype of *LoNAC29* OE and super vector control samples. (**B**,**C**) Expression of *LoNAC29* and ion leakage rate in *LoNAC29* OE and super control samples. The data represent the mean ± SE of three biological replicates. ** *p* < 0.01 (Student’s *t*-test).

**Figure 8 genes-12-00869-f008:**
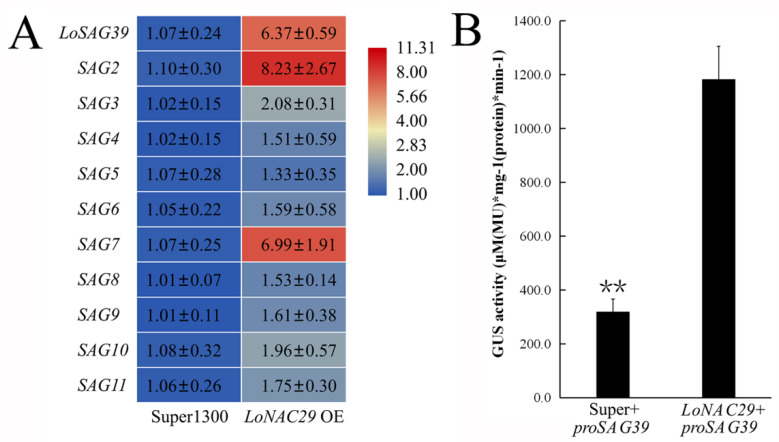
Regulation of senescence-associated genes (*SAGs*) by LoNAC29. (**A**) Expression of *SAG* genes in *LoNAC29* OE and super vector control samples. The color indicates the expression level of genes, and the data represent the mean ± SE of three biological replicates. *LoACTIN* was used as the internal reference gene. (**B**) LoNAC29 regulates the promoter activity of *LoSAG39*. The GUS reporter gene was used to indicate the promoter activity of *LoSAG39*. The data represent the mean ± SE of three biological replicates. ** *p* < 0.01 (Student’ s *t*-test).

**Table 1 genes-12-00869-t001:** Statistics of SMRT sequencing data before and after Cogent analysis.

Sample	Total_Nucleotides	Total_Number	Mean_Length	Min_Length	Max_Length	N50	Notes
lily_1_4	203,392,716	73,218	2778	145	14,462	3792	Before Cogent analysis
lily_1_4	186,674,771	62,960	2965	145	22,824	3935	After Cogent analysis

**Table 2 genes-12-00869-t002:** Full length statistics with TransDecoder analysis.

Sample	Total_Number	Full Length	5prime_Partial	3prime_Partial	Internal and Others
before Cogent	73,218	43,998 (60%)	12,938 (18%)	699 (1%)	15,583 (21%)
after Cogent	62,960	42,027 (67%)	7774 (12%)	548 (1%)	12,611 (20%)

## Data Availability

Sequencing data has been deposited in the National Center for Biotechnology Information (NCBI) Sequence Read Archive (SRA) (accession number PRJNA700112 and PRJNA700571).

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
