# Peer review of "SMRT and Illumina RNA Sequencing and Characterization of a Key NAC Gene LoNAC29 during the Flower Senescence in Lilium oriental ‘Siberia’"

_genes, 2021, doi:10.3390/genes12060869_

Round 1
Reviewer 1 Report
The authors studied the transcriptomic changes during the flower senescence of lily using the third-generation sequencing plus Illumina sequencing. Though there was no reference genome, they found thousands of DEGs and identified several genes that might play important roles in flower senescence. This study made a good example for future investigations in species without reference genomes. Nevertheless, there were still some issues remaining.
Major:
- Cultivar: There are many cultivars of L. oriental, the authors should explain why this 'Siberia' was chosen to be studied. Actually, Ref 17-19 chose three different cultivars including 'Siberia'. Are these cultivars very close in the respect of breeding? Even if they are close, there might still be SNP or InDels. How are these differences handled during the cogent analysis?
- Sampling: Is there any reference for the development stage of lily flower? The authors should make clear definitions on these stages. For example, stage 1 was defined as 'flowers are unopened, the buds are green'. But how large is the bud entering stage 1? We cannot state that because small buds were clearly not included here. Why do the authors collect the samples at 10am? For cut flower, why do the authors not cut from the lilies in the greenhouse which were used for sequencing?
- Parameters: For the entire methods about data processing, many references for software were missing and no detailed parameters were given. This obstructs repeating exprements in this studies by others.
- Known genes: During the DEG and TF analysis, were homologs to known genes in other species regulated in lily, such as RhHB1 and RhERF113?
- The authors stated that SMRT and Illumina sequencing together could improve the transcript annotation. However, no direct benchmark was made. The authors should add these numbers from the published data depending only on Illumina sequencing.
Minor:
- Line 304. The "5'_partial" should be the 5' end of the whole transcript, which means it's missing the 3' end.
- Figure 4. Should not use different scales showing qPCR results in one figure with only one color key.
- Line 531. This demonstration is way too strong. Co-expression does not grant a functional relationship. And similar conclusions were made several times. The authors should lower the tone.
Author Response
Dear reviewer, thank you very much for your suggestions. The point-to-point responses are listed in the attched file.

Reviewer 2 Report
I enjoyed the opportunity to read this interesting manuscript on comparative transcriptomics of genes involved in floral senescence in Lilium. The authors did a nice job of comparing particular floral stages to understand various ways in which flowers will age, although I'm not sure that some of the comparisons are necessarily the most appropriate (e.g., stage 1 vs 4). I thought the additional work on overexpression was helpful in gaining a deeper knowledge of the impact of these genes, but I should state that this is not my area of expertise. I believe there are many worthwhile aspects of the manuscript, but there are a few issues that should be explained in greater detail (outlined below, with line numbers where applicable). This is particularly the case for the methods, where I think that more information on parameters and data analysis would be helpful, and it would be nice to see more in the discussion related to the study as there is a great deal of work that seems to have gone into the submitted study.
-Data is plural.
-Scientific names should be italicized.
-I would suggest not using contractions.
127-In the methods, -80 C would be freezer, not a refrigerator.
127-How was the RNA and DNA isolated?
-It would be helpful to have more information on the methods related to the assembly of consensus sequences and the parameters for the DEG study.
-What was input into DESeq? It's unclear from the methods, but it seems like FPKMs. I don't think that FPKMs are the appropriate input for DESeq. This should be double checked by the authors, but according to this tutorial (http://www.bioconductor.org/packages/release/bioc/vignettes/DESeq2/inst/doc/DESeq2.html) raw counts should be used.
231-Twelve mm can be 12 mm.
-Tables 1 and 2 can probably be combined.
387-Features are homologous or not. I think that the authors are referring to sequence similarity.
-Were differences observed between the two horticultural varieties of lily?
-I think that the authors collected a lot of data, but the discussion seems like a missed opportunity to delve more into their results and to think not just about changes in expression, but the extent of these differences, and to explore some of the genes that perhaps hadn't been known to be involved in senescence previously and hypotheses as to the reason this may be the case. Additionally, if researchers were trying to breed lily flowers that would last longer, what would be the first few genes that researchers should examine. Also, are there differences in patterns of expression between the different lilies in the study
-I think there might be a better way to label the Venn Diagram in S3. I would expect the ellipses to be for each developmental stage, not for inter-stage comparisons.
Author Response

(The authors gave the same response as above.)

Round 2
Reviewer 1 Report
I was quite satisfied that the authors had explained my concerns and modified the manuscript according. There is only one point left. The authors should add the homologs to known genes to the results, which was raised as major point 4. These results would make the data analysis more convincing and grant direct access to data to people who are interested in functional studies.
Author Response
Dear reviewer, thank you very much for your suggestions. The responses are as follows.
In this work, we got 830 differentially expressed transcription factor genes, which were listed in Table S6. According to your suggestion, we searched the homologous genes in our SMRT sequencing data with published transcription factors which were reported to be involved in leaf or flower senescence, and found 13 NAC unigenes, as well as 1 HB-HD-ZIP unigene and 1 ERF unigene were homologous to ANAC029 (AtNAP) (Guo and Gan, 2006, Ref 8), ANAC046 (Oda-Yamamizo et al., 2016, Ref 33), ANAC032 (Mahmood et al., 2016, Ref 34), ANAC087 (Yan et al., 2018, Ref 35), ANAC022 (NAC1) (Xie et al., 2000, Ref 36), ANAC002 (ATAF1) (Takasaki et al., 2015, Ref 37), Ipomoea nil EPHEMERAL 1 (NAC092) (Shibuya et al., 2018, Ref 9), RhHB1 (Lü et al., 2014, Ref 10), and RhERF113 (Khaskheli et al., 2018, Ref 11) (Table S7), and their expression were significantly up-regualted at stage 4, suggesting that they might be important candidate transcription factors that regulates flower senescence in lilies. This part was added in the results as line 348 to line 354.
Reviewer 2 Report
Thank you for updating the manuscript. I think the manuscript has been improved by the changes. The authors seem to have used FPKMs as input to DEseq, which does not seem to be the appropriate input. The appropriate input, based on my understanding, is raw count data. If the wrong input is used, incorrect patterns of differential expression may occur. I mentioned this in the review, and I did not find the response satisfactory. I would recommend that this be clarified.Additionally, other more comprehensive aspects of my first review still stand, such as the discussion being a missed opportunity to expand on their results, which are certainly plentiful.
Author Response
Dear reviewer, thank you very much for your suggestions. The responses are as follows.
1. Differentially expressed geneswere analyzed with original read counts by using DEseq, sorry for the unclear description in the old versions. And the raw count data and analysis of differentially expressed genesbetween different flower opening stages were listed in Table S5.
2. In the discussion, we further analyzed the senescence-associated genes (SAGs) and key regulator during flower senescence. In this work, we got 319 differentially expressed genes which were highly expressed at stage 4, and 11 selected SAGunigenes were compared to SAG genes in Arabidopsis, and they were classified into three groups, SAG1(LoSAG39), SAG2 and SAG7 were homologous to the important senescence marker gene SAG12 in Arabidopsis, which encodes a cysteine protease (Guo and Gan, 2006, Ref8). SAG3, SAG10 and SAG11 were homology of AT1G11190, a bifunctional nuclease that acts on nucleic acid degradation, and the gene expression was extremely high in petal and stamen during flower senescence of Arabidopsis. What’s more, the sequence of KDEL cysteine protease unigenes SAG4, SAG5, SAG6, SAG8 and SAG9 were similar to AT5G50260 (AtCEP1), which was involved in pathogen defense. These work suggested that the 11 selected SAG unigenes might be involved protein/ nucleic acid degradation and pathogen defense during flower senescence in lilies. Except for the 11 SAG genes, we also discussed several hormone synthesis and signaling-related genes and F-box/ E3 ubiquitin ligase genes, suggesting the hormone signaling and E3 ubiquitin-protein ligase mediated protein degradation were widely involved in flower senescence of lilies. This part was added in the discussion as line 513 to line 541.
In addition, we also compared the differentially expressed transcription factors with published transcription factors which were reported to be involved in leaf or flower senescence, and found 13 NAC unigenes, as well as 1 HB-HD-ZIP unigene and 1 ERF unigene were homologous to ANAC029 (AtNAP) (Guo and Gan, 2006, Ref 8), ANAC046 (Oda-Yamamizo et al., 2016, Ref 33), ANAC032 (Mahmood et al., 2016, Ref 34), ANAC087 (Yan et al., 2018, Ref 35), ANAC022 (NAC1) (Xie et al., 2000, Ref 36), ANAC002 (ATAF1) (Takasaki et al., 2015, Ref 37), Ipomoea nil EPHEMERAL 1 (NAC092) (Shibuya et al., 2018, Ref 9), RhHB1 (Lü et al., 2014, Ref 10), and RhERF113 (Khaskheli et al., 2018, Ref 11)(listed in Table S7), and their expression were significantly up-regualted at stage 4, suggesting that they might be important candidate transcription factors that regulates flower senescence in lilies. This part was added in the discussion as line 560 to line 567.
Further more, we discussed the relation of chlorophyll metablism, synthesis of aroma components and flower senescence, which were added in the discussion as line 594 to line 598, and line 609 to line 620.